# Spatiotemporal analysis of the first wave of COVID-19 hospitalisations in Birmingham, UK

Samuel I Watson ,[1] Peter J Diggle,[2,3] Michael G Chipeta,[4] Richard J Lilford[1]

[1]Institute of Applied Health Research, University of Birmingham, Birmingham, UK
[2]Faculty of Health and Medicine, Lancaster University, Lancaster, UK
[3]Epidemiology and Population Health, University of Liverpool, Liverpool, UK
[4]Big Data Institute, University of Oxford, Oxford, UK

**Correspondence to**
Dr Samuel I Watson;
s.i.watson@bham.ac.uk

## ABSTRACT

**Objectives** To evaluate the spatiotemporal distribution of the incidence of COVID-19 hospitalisations in Birmingham, UK during the first wave of the pandemic to support the design of public health disease control policies.

**Design** A geospatial statistical model was estimated as part of a real-time disease surveillance system to predict local daily incidence of COVID-19.

**Participants** All hospitalisations for COVID-19 to University Hospitals Birmingham NHS Foundation Trust between 1 February 2020 and 30 September 2020.

**Outcome measures** Predictions of the incidence and cumulative incidence of COVID-19 hospitalisations in local areas, its weekly change and identification of predictive covariates.

**Results** Peak hospitalisations occurred in the first and second weeks of April 2020 with significant variation in incidence and incidence rate ratios across the city. Population age, ethnicity and socioeconomic deprivation were strong predictors of local incidence. Hospitalisations demonstrated strong day of the week effects with fewer hospitalisations (10%–20% less) at the weekend. There was low temporal correlation in unexplained variance. By day 50 at the end of the first lockdown period, the top 2.5% of small areas had experienced five times as many cases per 10 000 population as the bottom 2.5%.

**Conclusions** Local demographic factors were strong predictors of relative levels of incidence and can be used to target local areas for disease control measures. The real-time disease surveillance system provides a useful complement to other surveillance approaches by producing real-time, quantitative and probabilistic summaries of key outcomes at fine spatial resolution to inform disease control programmes.

## INTRODUCTION

A range of preventative public health measures have been deployed to limit the transmission of COVID-19 during the pandemic in 2020 and beyond. Perhaps most prominently 'lockdowns' of whole regions or nations, involving the closure of businesses and public spaces and requiring people to stay home, have been implemented worldwide. Lockdowns have been effective, but are blunt instruments, with potentially large collateral social and economic effects.[1] More localised approaches that target high-risk

## STRENGTHS AND LIMITATIONS OF THIS STUDY

⇒ Geospatial statistical methods can provide daily predictions of disease epidemiology at small spatial scales as a complement to large-scale designed studies that report infrequently.
⇒ Many countries and local authorities possess the data necessary to run real-time surveillance software that can readily be deployed on a desktop computer, but careful consideration of key model parameters is required.
⇒ Understanding the key predictors and extent of spatial and temporal correlation of disease incidence and transmission can help target local case area-targeted intervention programmes.
⇒ We report the output of a proof-of-principle surveillance system using geospatial statistical methods to predict local, daily COVID-19 risk using data on hospitalisations for COVID-19 in Birmingham, UK during the first wave of the pandemic in 2020.

areas and emerging disease clusters could potentially ameliorate some of these social and economic effects while maintaining low incidence rates, as has been demonstrated for other infectious diseases. For example, Ratnayake *et al* review the use of case area-targeted intervention (CATI) for cholera, which entails targeting a range of measures at small areas (50–100 m), including chemoprophylaxis, water treatment and vaccination, in response to the early detection of an outbreak.[2] However, the effective use of CATI requires both reliable early cluster detection and a good understanding of the spatial and temporal distribution of cases and clusters in order to delineate 'case-areas'.

There are a growing number of explicit or direct geospatial statistical analyses of COVID-19 transmission and spread.[3] Many of these examples use case data aggregated to small area levels (such as local government authorities or provinces)[4 5 3] and so cannot provide insight into the variability of COVID-19 incidence at more local levels, across urban areas. Therefore, there is almost

no published evidence or existing reporting systems that can guide CATI-type approaches for COVID-19. There are many 'indirect analyses' that compare local or larger scale risk factors with COVID-19 epidemiology, which could be used to identify high-risk areas for disease control intervention.[3 6 7] For example, population age has consistently emerged as a key risk factor in a number of studies.[8] In high-income countries, more socioeconomically deprived areas and areas with a high minority ethnic population proportion have also been identified as high risk.[9] But it is not known just how well these factors explain small-scale spatiotemporal variance in incidence rates.

Geospatial statistical analysis is increasingly in use as a tool for disease surveillance, mapping and epidemiology.[10] Incident cases and disease testing occur at discrete locations in space and time. Where data on these outcomes are spatially referenced and time stamped, they can be modelled to provide predictions of prevalence or incidence over an area of interest. Models can incorporate relevant covariate data and allow for spatial and temporal correlation, so that predictions are 'smoothed' over time and space. One specific application of these methods is for real-time surveillance.[11] Daily incident case data can be used to predict incidence rates across an area of interest, such as a city, in 'real time' to identify areas with a high probability of high or rising risk.[12] We developed a proof-of-principle real-time surveillance system using individual level data on COVID-19 hospitalisations in Birmingham, UK, although the software and statistical approach could be used for any disease. The system is designed to rerun the analyses with each new day's case data, thus its output represents an evolving spatiotemporal analysis of COVID-19 hospitalisations over the course of the pandemic. This article presents results from this analysis for the first wave of the COVID-19 pandemic.

## METHODS

The analyses presented in this article were conducted as part of a project to develop software for real-time disease surveillance. The aim was to develop a proof-of-principle system that could be deployed for COVID-19 surveillance, or for any disease where there exists spatially referenced and time-stamped individual-level case data. Most data sources available in 'real time' are positive-case (or proxy) outcomes from healthcare system databases, such as hospitalisations, presentations to health services or contact with public health telephone or internet services (such as NHS Direct in the UK). This contrasts with binomial testing data from large-scale surveys that occur on a less frequent basis[13]. The advantage of healthcare system data is that it also usually has patient residential address to enable geolocation of each case. Following previous applications of real-time surveillance systems,[11] we took a geospatial statistical approach to the development of the software and used data on COVID-19 hospitalisations to demonstrate proof-of-principle. The method we describe below

was primarily designed and set up to provide reliable predictions of incidence rates, and not for the purposes of estimating the effects of covariates or the nature of spatial and temporal correlation structure. However, outputs from these models provide useful descriptions of disease risk, its variation across an urban area and its correlations with key predictors. Our software is available online.[14]

## Data

We obtained data on all COVID-19 hospitalisations to hospitals in the University Hospitals Birmingham NHS Foundation Trust (UHB-FT) for the period 1 February 2020 to 30 September 2020, which covers the first wave of COVID-19 in the UK. A COVID-19 hospitalisation was defined as any inpatient admission, where the primary diagnosis was COVID-19; we did not specify how the diagnosis was made. Our primary data for these patients comprised their residential address and date of admission.

We defined our area of interest as the approximate catchment area of UHB-FT within Birmingham, which encompasses approximately 70% of the area of the city (figure 1). Cases resident outside this area were, therefore, excluded. We conducted the analysis as if the data were provided daily (see below), and each day's analysis included data from the preceding 14 days. Our first day was taken to be 23 March 2020, when the UK entered its first nationwide lockdown, so any cases admitted prior to 9 March 2020 were excluded from the data. Prior to 23 March 2020, there were few admissions overall, and most days had zero admissions. Residential addresses were converted into longitude and latitude coordinates using the Google Maps Application Programming Interface (API).

We obtained population and demographic data for lower layer super output areas (LSOA) covering the area of interest from the Office for National Statistics.[15] Each LSOA covers an average population of 700 people; there were 639 LSOAs within our area of interest. For each LSOA, we extracted and compiled projected 2019 population density (people per hectare), the proportion of the population aged over 65, the proportion of the population identifying as white ethnicity, the Index of Multiple Deprivation (IMD), an index used to capture socioeconomic deprivation based on outcomes including education, employment and crime.[16] The IMD is an ordinal index, and LSOAs are grouped into deciles, which we used in our analysis. We also determined from the date of admission, the day of the week of the case, which was also used as a covariate in the predictive model.

## Statistical model and computational methods

The Appendix gives a more detailed description of our statistical model and associated methods of inference. Here, we give an informal summary.

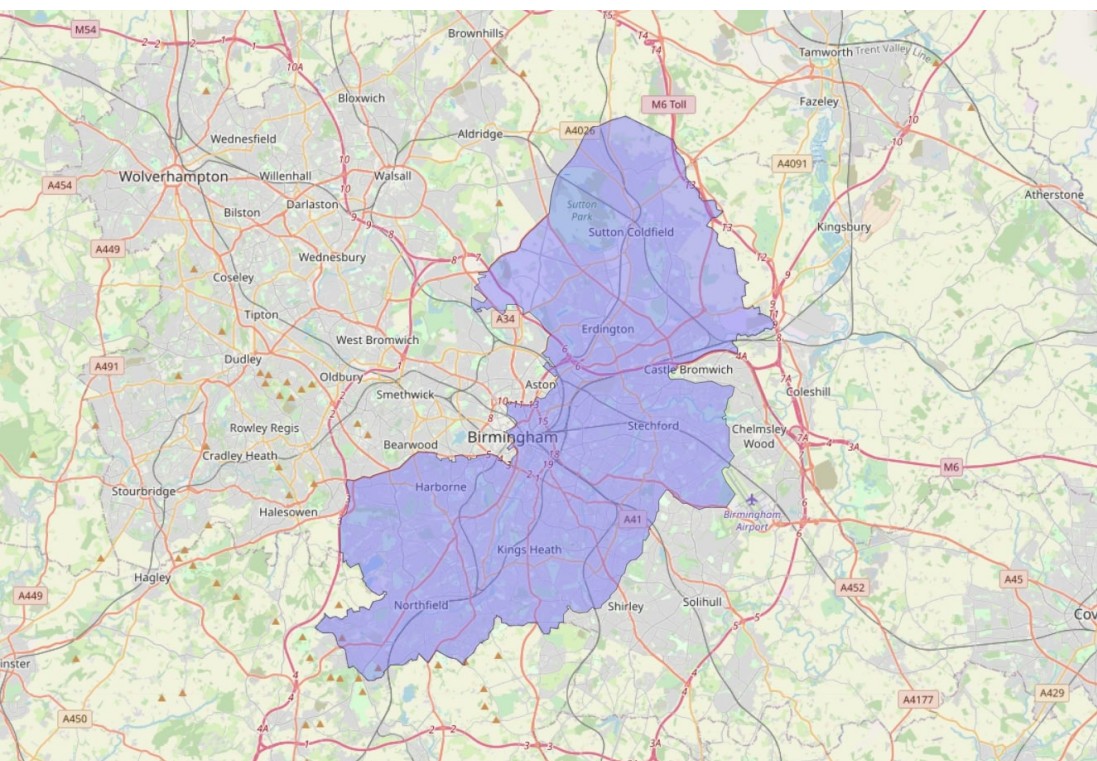

**Figure 1** Boundary of the area of study representing the approximate catchment area of University Hospitals Birmingham NHS Foundation Trust in Birmingham, UK.

Our statistical model is a spatiotemporal, log-Gaussian Cox process.[17] This model considers the study region as a spatiotemporal continuum, within which case incidence at location $s$ and time $t$ is expressed as

$$\lambda\left(s,t\right) = e\left(s\right)\exp\left(\beta_0 + \beta_1 x\left(s,t\right) + Z\left(s,t\right)\right).\# \qquad (1)$$

In equation (1), the terms within the exponential decompose the spatiotemporal variation of individual-level risk into three multiplicative components, a constant of proportionality $\exp\left(\beta_0\right)$ and two varying terms $\exp\left(\beta_1 x\left(s,t\right)\right)$ and $\exp\left(Z\left(s,t\right)\right)$. The first of these is a log-linear regression that accounts for variation in risk that can be explained by measured characteristics $x\left(s,t\right)$. The second accounts for any residual, unexplained variation, which we represent as the unobserved realisation of a stochastic process. Specifically, we assume that $Z\left(s,t\right)$ is a spatially and temporally correlated Gaussian process. The term $e\left(s\right)$ is the population density (number per hectare) at $s$, which converts risk into an expected incidence per hectare per unit time; we assume that $e\left(s\right)$ does not change materially over the time window of the data.

To fit the model, we use Bayesian inference to obtain the joint predictive distribution of $\lambda\left(s,t\right)$ at all locations $s$ and time $t$, given all available data up to and including time $t$. This allows us to calculate and display suitable summaries of the predictive distribution for whatever properties of the complete history of the incidence surface we wish, for example, current incidence; change in incidence from 1 day (or week) to the next; exceedance of a prespecified incidence threshold;

decomposition of incidence into explained and unexplained components. This flexibility is a crucial aspect of our approach, which allows predictions to be made in whatever form is most relevant to each user's needs.

Although our model is formulated in a spatiotemporal continuum, for computation, we approximate the study region by a grid of square cells with side length approximately 50 m, and record the time of each case as an integer number of days since 23 March 2020.

### Presentation of results

The model described in equation (1) can be used to generate a range of outputs and predictions relevant to disease surveillance for each lattice cell. These include:

1. Incidence of COVID-19 hospitalisations per 10 000 person-days. We notate this as $I_{st}$ for grid cell $s$ at time $t$.
2. The relative risk of each cell associated with the observed covariates, 'observed RR': $\exp\left(\beta_1 x\left(s,t\right)\right)$ in equation (1).
3. The relative risk associated with the Gaussian process 'Latent RR': $\exp\left(Z\left(s,t\right)\right)$ in equation (1). This component captures variation in risk associated with unobserved or unexplained local factors.
4. The incidence rate ratio (IRR). For each grid cell, we determine the incidence per 10 000 person-days relative to the same location 7 days prior: $I_{st}/I_{s,t-7}$
5. The cumulative incidence up to time $T$, which is $\sum_{t=1}^{T} I_{st}$.

We graphically examine the posterior means of these outputs for the grid cells. We also report the posterior

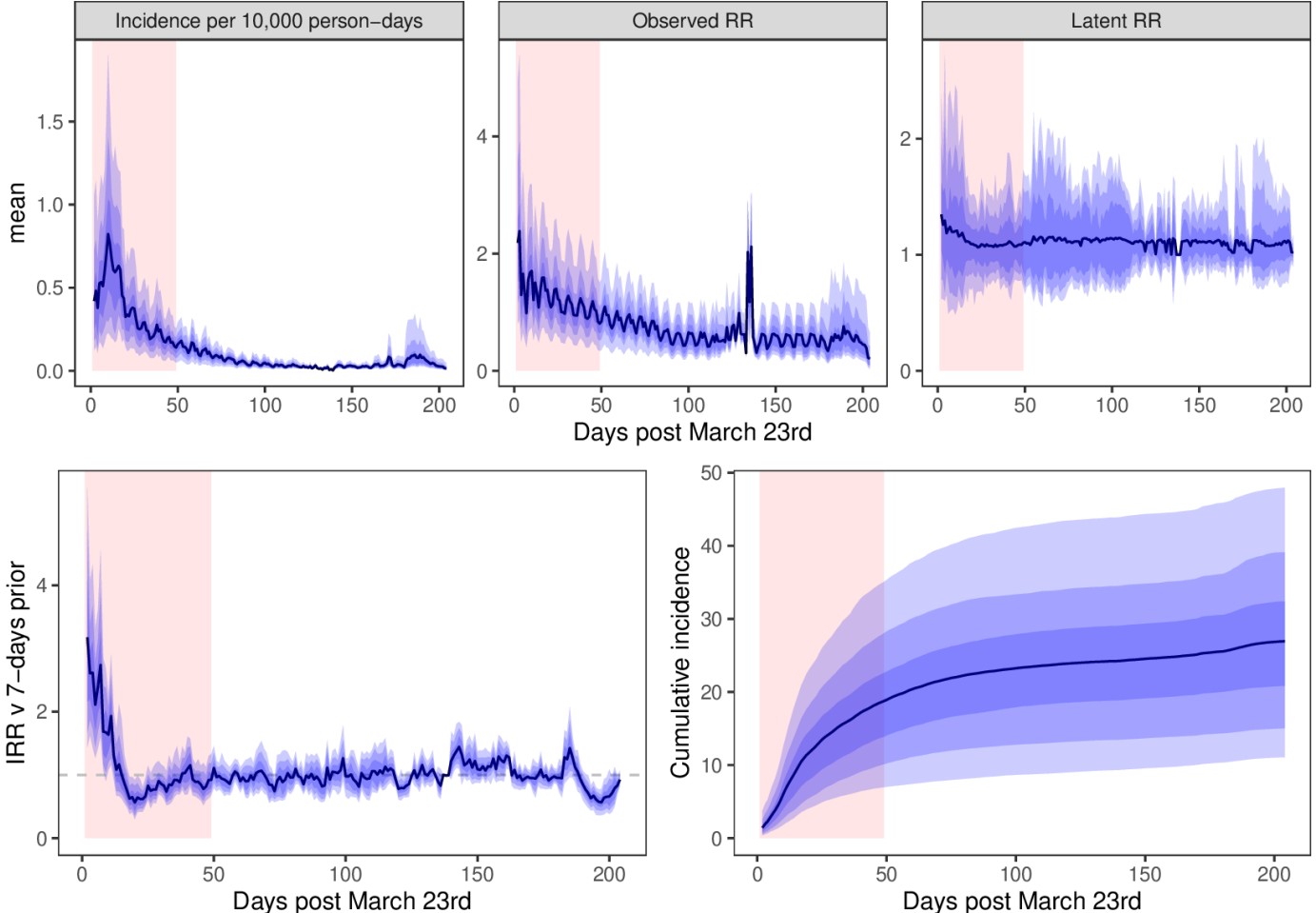

**Figure 2** Model predictions of city-wide COVID-19 incidence, relative risks, 7-day incidence rate ratio (IRR) and cumulative incidence. Plots show mean value of posterior mean predictions across the city with bands for 50%, 80% and 95% intervals. The March–May lockdown period is highlighted in red.

distributions for the model parameters (both in the linear predictor and covariance function) at different points during the first wave (1 April, 1 May and 1 June 2020).

Finally, we also consider the 'persistency' of risk across the area of interest. For each day of the analysis, we identified the top 10% of lattice cells in terms of population standardised incidence. For each cell, we then determined the proportion of days it was in the top 10%. If risk was completely random across the area, then each cell should appear in the top 10% approximately 10% of the time. If it was completely deterministic, then 10% of cells would appear 100% of the time.

## RESULTS

For the period 23 March to 30 September, there were 4040 recorded admissions for COVID-19 to UHB-FT hospitals. Of these, 2668 could be geolocated and were resident in the area of interest. Figure 2 summarises the model predictions. Figure 3 provides model geographic outputs for 1 April 2020.

The first wave of the pandemic is evident from days 1 to 50 after March 23, with peak hospitalisations occurring

in the first and second weeks of April 2020. A day-of-the-week effect is evident in the incidence, particularly considering the observed relative risk. Table 1 reports estimates of model parameters; the rate of hospitalisations was 15%–30% lower at the weekend than on weekdays. There was significant variation in incidence rates across the city. For example, the 2.5th and 97.5th quantiles of posterior mean incidence rates on March 30 (day 7) differed by a factor of approximately 10, they were 0.14 and 1.11 hospitalisations per 10 000 person-days respectively. This difference is also reflected in the cumulative incidence: by day 50 at the end of the first lockdown period the top 2.5% of cells had experienced five times as many cases per 10 000 population (>35) as the bottom 2.5% (<7).

Figure 2 also shows the 7-day IRR, for which both the mean and median posterior mean remained greater than 1 until day 15 of the lockdown. By day 18, the posterior mean was lower than one for almost all cells. Figure 3 shows that there was significant variation in the IRR across the city – in the north of the city one small area had both relatively high predicted incidence and high IRR. A sudden 'spike' in observed relative risk is observed

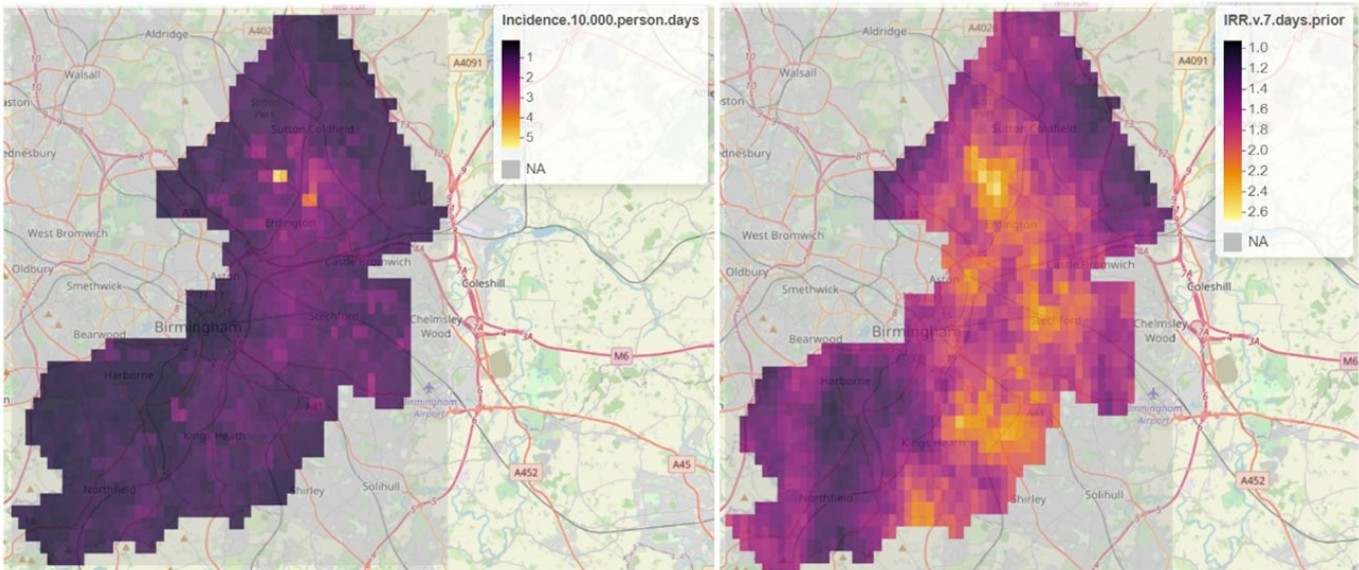

**Figure 3** A sample of surveillance system outputs for 1 April 2020. Left: predicted incidence of COVID-19 hospitalisations; right: incidence rate ratio (IRR) relative to 7 days prior.

around day 140 when hospitalisation rates approached zero.

Population age, ethnicity, and socioeconomic deprivation were all strongly associated with COVID-19 hospitalisation rates (table 1). The effect of age was largest in magnitude: a 10 percentage point (pp) increase in the proportion of the population aged over 65 was associated with an approximate doubling in the rate of hospitalisation. A 10 pp increase in the proportion of the population identifying as white was associated with an approximate 10% reduction in incidence. Good evidence

of a 'weekend effect' was also apparent; patients were less likely (~10%–20%) to be admitted at the weekend than at the weekday.

As a heuristic, one can multiply the spatial and temporal range parameters by three to get approximate upper limits of the distance and time over which observations are correlated (see online supplemental appendix 1 for further statistical details). The spatial correlation was approximately of the order of 0.2 km—unexplained increases in incidence were similar across areas of this magnitude. However, the estimated temporal range was

**Table 1** Posterior mean (95% credible intervals) of model parameters from analyses conducted on three dates

| Parameter | April 1 | May 1 | June 1 |
|---|---|---|---|
| *Linear predictor* | | | |
| 10 pp increase in proportion of population aged 65 and over | 2.12 (1.96, 2.29) | 1.86 (1.81, 1.91) | 1.72 (1.66, 1.78) |
| 10 pp increase in proportion of population identifying as white ethnicity | 0.92 (0.90, 0.94) | 0.92 (0.91, 0.92) | 0.91 (0.90, 0.92) |
| Increase (less deprived) in IMD decile (linear) | 0.93 (0.91,0.95) | 0.91 (0.90, 0.92) | 0.89 (0.88, 0.90) |
| Monday | 0.99 (0.86, 1.15) | 0.85 (0.82, 0.89) | 0.98 (0.91, 1.06) |
| Tuesday | 1.10 (0.94, 1.26) | 1.01 (0.96, 1.07) | 1.09 (1.02, 1.17) |
| Wednesday | 1.12 (0.95, 1.31) | 1.12 (1.08, 1.17) | 1.11 (1.05, 1.17) |
| Thursday | 0.73 (0.62, 0.86) | 1.05 (0.99, 1.12) | 0.92 (0.87, 0.97) |
| Friday | Ref. | Ref. | Ref. |
| Saturday | 0.84 (0.73, 0.80) | 0.84 (0.80, 0.89) | 0.85 (0.80, 0.90) |
| Sunday | 0.68 (0.58, 0.80) | 0.83 (0.79, 0.88) | 0.75 (0.71, 0.79) |
| *Covariance parameters* | | | |
| Sigma | 1.12 (1.05, 1.19) | 1.05 (1.04, 1.06) | 1.05 (1.04, 1.06) |
| Spatial range | 0.07 (0.07, 0.08) | 0.07 (0.07, 0.08) | 0.08 (0.07, 0.08) |
| Temporal range | 0.16 (0.13, 0.20) | 0.11 (0.10, 0.11) | 0.10 (0.09, 0.11) |

IMD, Index of Multiple Deprivation.

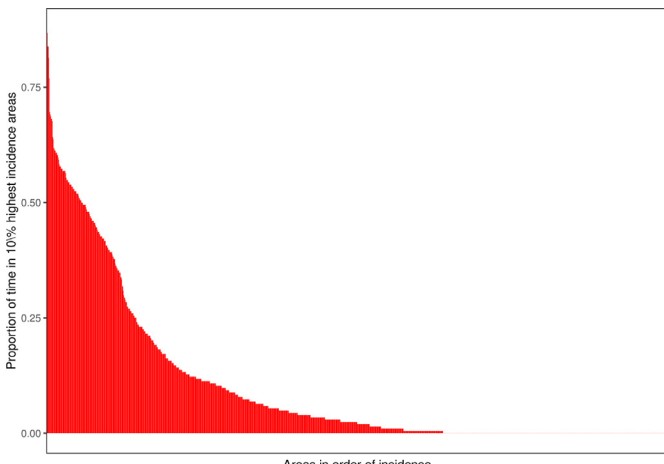

**Figure 4** Proportion of time each grid cell was in the highest incidence 10% of grid cells.

substantially shorter than a day. This implies that the unexplained component of local incidence on any 1 day, $x(s,t)$, is poorly predictive of the unexplained component on the next day. Given the lack of temporal correlation, one might ask whether it is possible to predict the location of local disease clusters given previous data. Figure 4 0 reports the 'persistence' of risk for the 1411 grid cells across the city. Four grid cells were in the top 10% of grid cells by daily incidence more than 75% of the time. Approximately, 5% of grid cells were in the top 10% of grid cells by incidence more than 50% of the time. Approximately, 40% of grid cells were never in the top 10%. These results suggest that one could predict a high-risk area with reasonable confidence based on its observed characteristics $x(s,t)$.

## DISCUSSION

Our results agree with observations made early in the COVID-19 pandemic, namely, that age and ethnic minority status are associated with increased risk of COVID-19 transmission.[8 9] Indeed, these factors along with a general measure of socioeconomic deprivation could reasonably predict the small areas of Birmingham with a relatively high incidence of COVID-19 hospitalisations. However, further research is required to determine whether these associations persist in subsequent disease waves to guide longer term public health policy.

We estimated a low temporal correlation in unexplained variation in incidence. This is likely due to the attenuation effect of identifying cases by their date of admission rather than by their (unknown) date of infection, which could have occurred several days or weeks before admission. This is a limitation of many sources of real-time surveillance data, including positive test data. Nevertheless, these results also indicate the strength of our underlying statistical approach, as it enables us to precisely quantify the degree of certainty, or lack thereof, associated with any predictions we make about local disease risk. Crude positive test counts have been the

most reported statistic for disease surveillance during the COVID-19 pandemic in the UK and elsewhere.[18] These counts are correlated with the underlying incidence and can provide useful and rapid feedback to officials to implement disease control methods such as lockdowns and tracking and tracing programmes. However, they are a blunt instrument as they do not reflect uncertainty in the underlying estimates of incidence, nor can they offer predictions of locally varying relative risk and associated risk factors. Inferences based only on crude outcomes can easily lead to the identification of signals where none exist.

The results in this article lead to some conclusions to support policy, both specifically for COVID-19 and for disease surveillance in general. First, at the time of writing, vaccination programmes are rolling out in the UK and elsewhere. A spatial approach should be taken with respect to prioritising who gets the vaccination. Transmission occurs locally, and areas with older populations, more ethnic minority residents, and that are more socioeconomically deprived have higher rates of transmission. These areas should be prioritised. Second, predicting small-scale day-to-day changes in incidence is difficult and highly uncertain, especially once a pandemic has reached an exponential growth phase. Real-time surveillance systems may have the greatest utility for monitoring diseases that are under general control in the population to enable rapid, targeted responses to emergent disease clusters. Once transmission levels are high across the population, there are no longer isolated clusters, although measured characteristics can still give reliable predictions of local relative transmission rates. Third, there are several scientific and statistical approaches to disease surveillance.[17 19] Epidemiological transmission models provide a framework for modelling and estimating transmission rates,[20] but currently available models cannot easily incorporate locally varying and spatially correlated effects. Moreover, these models often require high-quality data, for example, from randomised studies, that can only be collected at relatively infrequent intervals. Geospatial statistical models fitted to electronic health record data offer an alternative method of quantifying uncertainty and making real-time probabilistic predictions that complement other approaches. The methods described in this article use publicly available software and can be run on a desktop machine in a matter of hours using routinely collected data. Thus, there is a strong case for incorporating them into the suite of public health tools used for disease surveillance.

We note some weaknesses of our approach. We have cautiously interpreted the parameters associated with local covariates, including age and ethnicity, but all estimates in spatial models are biased to some extent.[21 22] We excluded the data from many admissions, principally because they were out of area, however, a significant proportion could not be geolocated. If the missingness of location data were correlated with latent disease risk then this would prejudice our predictions. However, we

believe such correlations are likely small. Data quality is one of the key limitations of using routine health system data across all applications, so results from large, high-quality studies should be used to validate any results where possible. Extensions to the type of model used in this article may provide a better fit to the types of data we describe, for example, a Hawkes process model allows for new cases to increase the probability of subsequent cases. However, the greater complexity of such models may require increased computational resources and so limit their usefulness for the types of applications described here. Similarly, alternative approximations may provide a more desirable trade-off between computational time and accuracy.[23] Further research in this area is warranted.

There have been previous examples of proof-of-principle real-time surveillance systems. For example, the Ascertainment and Enhancement of Gastrointestinal Infection Surveillance andStatistics (AEGISS) project used daily data from the then NHS Direct telephone service to identify potential outbreaks of gastrointestinal disease in Southern England.[11] However, there have been few well-documented implementations of such systems since despite advances in software and availability of computational resources. Localised spatiotemporal interventions have been trialled for several conditions, such as cholera, and have been broadly successful.[2] Real-time surveillance tools can provide an important and timely input into targeting them.

**Contributors** SIW, RJL, PD and MC conceived of the research idea. SIW led the development of the software, statistical analyses and manuscript writing, with support from RJL, PD and MC, who edited and approved the final version of the manuscript.

**Funding** UKRI/DHSC COV0036. The funders had no role in study design, data collection and analysis, decision to publish, or preparation of the manuscript.

**Map disclaimer** The inclusion of any map (including the depiction of any boundaries therein), or of any geographic or locational reference, does not imply the expression of any opinion whatsoever on the part of BMJ concerning the legal status of any country, territory, jurisdiction or area or of its authorities. Any such expression remains solely that of the relevant source and is not endorsed by BMJ. Maps are provided without any warranty of any kind, either express or implied.

**Competing interests** None declared.

**Patient consent for publication** Not applicable.

**Ethics approval** Ethical approval for use of the hospital admissions data presented in this article was granted by HRA and HCRW (IRAS 288478).

**Provenance and peer review** Not commissioned; externally peer reviewed.

**Data availability statement** No data are available. Data are not available for sharing from this project. All statistical code and documentation is publicly available at the links provided in the article.

**ORCID iD**
Samuel I Watson http://orcid.org/0000-0002-8972-769X

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
