## [Reviewer comments · BMJ Open]

ARTICLE DETAILS

TITLE (PROVISIONAL)	Spatio-temporal analysis of the first wave of Covid-19 hospitalisations in Birmingham, UK
AUTHORS	Watson, Samuel; Diggle, Peter; Chipeta, Michael; Lilford, RJ

VERSION 1 – REVIEW

REVIEWER	Ding, Yongmei Wuhan University of Science and Technology
REVIEW RETURNED	22-Apr-2021

GENERAL COMMENTS	In this paper, the authors develop a proof-of principle real-time surveillance system to analyze spatio-temporal characteristics of Covid-19 hospitalizations in Birmingham, which provides some guidance to vaccination programmes. From my point of view, the work is well-done and provides interesting results to the real-time surveillance systems, and it's meaningful to the prevention and control policy. thus it merits to be published. The subject considered is an interesting one and the paper is well organized. The Introduction gave a satisfactory literature survey on the similar topic and it outlined the proposed method well in the part of discussion. . I suggest it published for its timeliness asap.
--

REVIEWER	Briz-Redón, Álvaro City Council of Valencia, Statistics Office
REVIEW RETURNED	01-May-2021

GENERAL COMMENTS	In this paper, the authors have performed a spatio-temporal analysis of COVID-19 geocoded data, considering several covariate effects and a Gaussian process accounting for temporal and spatial correlation in the data. The content of the paper is relevant from both an epidemiological and a methodological perspective since many of the already published papers on this topic have been carried out at the area level. Besides, the paper is very well written and the statistical analysis is nicely explained. Therefore, I think that this paper must be published. I only have the following minor comments: - In the Results section, in the paragraph starting with "As a heuristic...", the authors discuss the estimates obtained for the parameters involved in the double-exponential correlation function that controls the covariance of the Gaussian process. I think that it is convenient, as the authors have done, to avoid further details on this correlation function in the main text, but an explicit reference to the Appendix at this point might be beneficial for the reader.
--

	- The authors state that their model might fail to predict local disease clusters. Have the authors considered using a Hawkes or self-exciting process instead of the Gaussian process with spatial and temporal autocorrelation structure? Some comments on this alternative modeling approach might enrich the discussion.
--	--

REVIEWER	Ramirez, Christina UCLA, Biostatistics
REVIEW RETURNED	11-Aug-2021

GENERAL COMMENTS	The authors implement a geospatial model to test if a CATI (Case Area Targeted Intervention), which has been used in other infectious diseases such as cholera, could be used for COVID-19. Use of geo-spatial data is a real strength of the article and offers a proof-of-concept that if one can detect clusters accurately then targeted interventions could reduce the necessity of a much more blunt tool, which is the lockdown. Prior studies used aggregated data from local areas and found that age and economic deprivation as well as areas with high ethnic minority populations are key predictors. The authors use data from the University Birmingham hospital system from March 23- September 30, 2020. The authors use patient addresses to get the location to see if it fell in the catchment area of UHB-FT which comprised approximately 70% of Birmingham. They then divided the area into blocks (square cells) to cover the catchment area. Hits resulted in 2,668 geolocated hospital admissions (out of 4,040). Then then applied a geospatial model as a log Gaussian Cox process, a well known and studied model. They record the time of each case as an integer number of days since March 23, 2020. Having estimates of uncertainty for these public health statistics such as incidence is a real strength. The persistency of risk is very interesting and indeed the authors establish their proof of concept in this small study that certain cells have persistent risk and is associated with known risk factors for COVID-19 spread, namely age and ethnicity and deprivation. The study seems reasonable and well done and the authors make their code available. Minor questions: The authors are using 66% of the total hospitalizations at the time. Are the 2,668 representative of hospitalizations in this area during this time? The question remains, is this generalizable across time and pandemic waves. That is, do we see the same grids at risk in subsequent waves. I realize this may be an unreasonable request for this manuscript, but it would go a long way to assess the generalizability of this method for making public health recommendations, if it is reproducible in subsequent waves.
---

VERSION 1 – AUTHOR RESPONSE

Comment	Response
Reviewer 1	
No response required	
Reviewer 2	
In the Results section, in the paragraph starting with “As a heuristic...”, the authors discuss the estimates obtained for the parameters involved in the double-exponential correlation function that controls the covariance of the Gaussian process. I think that it is convenient, as the authors have done, to avoid further details on this correlation function in the main text, but an explicit reference to the Appendix at this point might be beneficial for the reader.	We have added a reference to the appendix in this paragraph and a note in the appendix explaining this further.
The authors state that their model might fail to predict local disease clusters. Have the authors considered using a Hawkes or self-exciting process instead of the Gaussian process with spatial and temporal autocorrelation structure? Some comments on this alternative modeling approach might enrich the discussion.	We have added a paragraph to the discussion to highlight the potential benefits of other modelling methods such as the Hawkes process with the note that further research is required on application of these methods to real time surveillance.
Reviewer 3	
The authors are using 66% of the total hospitalizations at the time. Are the 2,668 representative of hospitalizations in this area during this time?	The majority of the excluded cases were excluded for being out of area so we are not too concerned if they are different, since we would expect differences by area. However, we have added a note that the lack of usable location information for cases in area may affect our predictions missingness is correlated with disease incidence risk.
The question remains, is this generalizable across time and pandemic waves. That is, do we see the same grids at risk in subsequent waves. I realize this may be an unreasonable request for this manuscript, but it would go a long way to assess the generalizability of this method for making public health recommendations, if it is reproducible in subsequent waves.	Unfortunately we do not know this as we do not have access to more recent data, and obtaining it would require several months of approvals. However, we have noted that further research is needed on the persistence of risk to guide public health efforts.

VERSION 2 – REVIEW

REVIEWER	Briz-Redón, Álvaro City Council of Valencia, Statistics Office
REVIEW RETURNED	01-Sep-2021

GENERAL COMMENTS	The authors have resolved my minor concerns. I think that the paper should be published in its current form.
--

REVIEWER	Ramirez, Christina UCLA, Biostatistics
REVIEW RETURNED	08-Sep-2021

GENERAL COMMENTS	The revised paper is responsive to the critiques.
---